# Investigating the Self-Perception of Social, Emotional, and Academic Inclusion of Students with and without Special Educational Needs through Photovoice

Alexandra Pirker, Julia Hafenscher and Katharina-Theresa Lindner *

Center for Teacher Education, University of Vienna, 1090 Vienna, Austria; alexandra.pirker@univie.ac.at (A.P.); julia.hafenscher@univie.ac.at (J.H.)
* Correspondence: katharina-theresa.lindner@univie.ac.at

**Abstract:** Several studies have investigated the perceptions of inclusion by students with special educational needs (SEN) and without SEN, most of them quantitatively. This research aims to expand the understanding of the perceived inclusion of students through qualitative interviews by examining how emotions, social relationships, and academic concepts matter. Therefore, the photovoice method was used for data collection, followed by semi-structured interviews, which were analyzed according to Mayring's Qualitative Content Analysis. Regarding social integration in the classroom, the results of the interviews with six students with SEN and three students without SEN show mainly positive experiences with their classmates and attach great importance to school spaces (e.g., the schoolyard) that are used for interaction and communication among each other. In terms of the teacher–student relationship, it becomes apparent that the students place particular value on the support and help of the teachers in everyday school life as well as in private matters. Some students' statements indicate that the self-perception of their academic self-concept differs in various school subjects. A contrast of perception between the students with and without SEN was not detected. Concerning emotional inclusion, the students primarily expressed statements related to emotion regulation and individual adaption strategies.

**Keywords:** perception of inclusion; students' self-perception; special educational needs; academic self-concept; socio-emotional inclusion; photovoice





## 1. Introduction

Inclusive education has been a common goal in Europe since at least the Salamanca Conference in 1994 and the ratification of the United Nations Convention on the Rights of Persons with Disabilities in 2006 (for an overview, see, e.g., [1]). In this study, inclusive education is defined as not only including students with and without special educational needs (SEN) in the same classes, but also focusing on equity in the students' academic and socio-emotional development [2]. An in-depth approach that investigated three different levels of inclusion, emotional, social, and academic, was utilized. This was in order to capture perceptions of inclusion not only at the SEN approval level, but also the thoughts and feelings of all participating students regarding their perceived inclusion.

### 1.1. Implementation of Inclusive Education on Different Levels

The task of inclusive education is to meet the individuality and needs of each learner and to prevent exclusion not only on the academic level, but also the social and emotional levels, unconditionally [3]. It is important to base inclusive education not only on the aspect of learning, but also on social inclusion, which refers to a broader social context and to an interpersonal context characterized by relationships and emotions [4], as well as ensuring one's emotional needs [5] are met. In contrast to non-inclusive schools, which

expect students to adapt to the existing school system, inclusive schools modify the school environment to cater to the individual needs of each student [6].

The student version of the Perceptions of Inclusion Questionnaire (PIQ) is a quantitative approach to measuring students' self-reported perceptions of inclusion consisting of 12 items. The questionnaire is freely available in different languages and measures inclusion in heterogeneous groups according to an inclusive approach [7]. Thereby, the PIQ addresses the three dimensions of emotional inclusion, social inclusion, and academic self-concept [8]. The approach of this study is to further investigate whether the three dimensions of the PIQ are also verifiable on a qualitative level, in this case by photovoice interviews with the participating students, and whether the three dimensions are autonomously named by the participants or whether only the sub-areas of the levels of inclusion based on the PIQ concept are mentioned, or none at all.

### 1.2. Emotional Inclusion in School

Previous research shows that, among other factors, students' perceived emotional inclusion is an important determinant of social participation in the school context [9]. There are multiple approaches to theoretically define the concept of students' emotional inclusion and ways to measure this concept [10]. In summary, perceived emotional inclusion can be described as the experience of positive feeling states toward the school itself, people in the school, and the context of school in general on a cognitive, emotional, and physical level [10,11]. Successful and effective learning is closely linked to emotional inclusion. Self-perceived emotional inclusion influences school engagement and, subsequently, successful learning based on attitudes toward the school attended. [12]. Inconsistent findings are observed when comparing the emotional inclusion of students with SEN to those without SEN in inclusive classes. It should be noted that SEN is a broad term and is defined and categorized variably in different countries based on diagnostic tests and criteria. In Austria, the country in which the present study was conducted, there is a distinction of the official diagnosis of SEN between sensory disabilities, physical disabilities, mental disabilities, and learning disabilities. This study focuses on SEN-L (special educational needs of learning), because the majority of the students in this study's sample can be assigned to this type of SEN.

Preliminary research reports lower emotional inclusion scores for students with SEN (all types of impairment included), e.g., through their attitudes toward and affiliation with school [13], or their emotional and social well-being in the context of school [14]. However, the study conducted by Schwab et al. [15], where mainly students with SEN-L were included, found no significant difference between the two groups. An important influencing factor for emotional inclusion is emotion regulation, as the socially appropriate handling of emotions is an important success factor for social relationships and belonging in general [16]. Considering that emotions have a major influence on human actions, behavior, and well-being, the regulation of emotions plays a crucial role. Based on this, people always strive to feel pleasant emotions and to eliminate, reduce, or avoid negative emotions that are perceived as unpleasant [17]. Therefore, emotion regulation must be considered when investigating emotional inclusion.

### 1.3. Social Inclusion in School

An important component of school well-being includes social participation [18]. Social participation can be defined as the level to which the students perceive themselves to be part of a particular group [19] and is considered a key element for the students' self-perceived inclusion [2]. A study of SEN students' (including mainly students with SEN-L) perceptions of factors that promote their social inclusion in inclusive classes shows that peers and classmates have a major influence on their perceptions of belonging to the class, followed by teachers and other educational staff. For students without SEN, the study reveals a common sense of togetherness as an important factor in their perception of inclusion [20]. Research consistently indicates that students with all types of SEN are

more likely to experience low social participation, which can impact their well-being (e.g., [21–24]). Nonetheless, some studies did not find any significant differences between the students with SEN and their peers regarding self-rated social participation [23]. The attribution that comes from attending a special school can lead to less experienced social inclusion not only within the school, but also outside this context, which in turn, highlights the complexity of inclusion at different levels. In fact, a study found that about half of the students with SEN reported bullying by neighbors or peers outside of school, whereas only a small proportion of the students without SEN reported the same [25].

In addition to social participation on the peer relationship level, the student–teacher relationship at school also plays an important role. Recent past research shows that the student–teacher relationship influences students' emotional and social development (for an overview, see, e.g., [26]). In the case of students with SEN, the majority were diagnosed with SEN-L; Schwab and Rossmann [27] found evidence that these student–teacher relationships were more unstable compared to those in the students without SEN. Long-term studies show that there is a connection between the poor quality of peer relationships and student–teacher relationships and the development of depression [28,29].

### 1.4. Academic Inclusion in School

Despite the strong focus of recent research on school achievement and academic performance in inclusive settings, following the PIQ [30], the academic self-concept can be seen as a primary indicator of determining students' self-perceived inclusion in school [20,31,32]. In addition to social and emotional inclusion, the construct of academic self-concept can be seen as an important component of the students' perceptions of successful inclusive education and academic inclusion [2]. The academic self-concept describes an individual's self-perception and expectations of their academic achievement, and subsequently influences their cognitive abilities. In turn, the academic self-concept is an important factor in students' academic and personal development [33,34]. Regarding the academic self-concept of students with and without intellectual disabilities, the systematic review of Maïano et al. [35] displays mixed results. Whereas two studies [36,37] report significant differences in students' cognitive–academic self-concept depending on whether they have an intellectual disability or not (higher level of cognitive self-concept among students without an intellectual disability), one study [38] did not find any differences between the student groups [35]. From the students' perspective, it was demonstrated that students with SEN who had already attended regular and special schools received less support from teachers in inclusive settings due to larger class sizes, and therefore, less staff support [39,40]. This underscores the importance of actual design in terms of resources in an inclusive school setting.

### 1.5. The Present Study

To gain a deeper insight into the everyday school life of students with and without SEN and to identify important topics regarding their perception of school on emotional, social, and academic inclusion, interviews were conducted using the photovoice method [41]. Due to the underlying claim of inclusion in this study, the participating students must be perceived as independent protagonists with legitimate feelings and thoughts from the child's point of view to depict their reality. The photovoice method has a participatory approach, which allows the students to shape the research results and to fill the photos in the interviews with their meaning and reality [42].

The data for this study were collected as part of the ongoing PATHWAY project. Due to this fact, the additional aim of the article is to identify important topics from the point of view of the students with SEN, which may need to be considered in further data collection. Therefore, the article investigates the following research questions:

- How do students with and without SEN-L (special educational needs in learning) display and perceive their own social, emotional, and academic inclusion in everyday school life?

- Are participants' perceptions of their school inclusion at the intrapersonal level similar across the three domains (social, emotional, academic inclusion), or are there differences in their perceptions of inclusion across the three domains?

## 2. Data and Methods

The data analyzed in the course of the current study are derived from the second measurement phase of the research project PATHWAY (academic and socio-emotional development of students with learning disabilities in their transitional phase from primary to lower-secondary school in inclusive education; funded by the Oesterreichische Nationalbank, funding number: 18567), which took place in September/October 2021. The PATHWAY project follows a mixed-methods design including both quantitative and qualitative data collection procedures encompassing different perspectives (students, teachers, parents). The specific objective of the research project is to provide insights into how learning environments (regular, inclusive, and special education classes) affect individual learning trajectories, educational biographies, and the socio-emotional development of students. The project is designed as a longitudinal study. In this context, the students attending school in Vienna, Austria are empirically followed from the fourth grade of primary school to their entrance into work life to trace their educational biographies as well as factors that promote and hinder positive educational experiences. Against this background, the main objective of the project deals with the socio-emotional and academic development of students with and without learning disabilities throughout their educational pathways.

### 2.1. Photovoice Study with Students with and without SEN

The method of photovoice developed by Wang and Burris [41] was utilized for data collection in this study. This research approach is commonly used for participants who have limited linguistic ability and allows for a deeper understanding of how students perceive certain things. Students were provided with printed emojis representing Ekman's six basic emotions including anger, disgust, fear, happiness, sadness, and surprise, for which a randomized sample of students voted in a pre-study [43]. The inclusion of peers in the selection process of the emojis, which were to represent the basic emotions, was intended to facilitate the age-specific reference and, in principle, the possibility for all participants to connect to the emotions represented [44]. Along with this approach, the researchers provided the students with information sheets and a task to take photos of places and objects on the school grounds that were meaningful to them concerning their perceived emotions. Researchers did not provide students with the intended meaning of the emojis, but rather gave students the possibility to interpret the pictures themselves. In addition to the photo collection, semi-structured one-on-one interviews were conducted with the participants to gain an in-depth understanding of their photographs and associated emotions.

Within the course of the semi-structured interviews, the main topics of discussion included (1) joint discussion of the photos to clarify the motive and emotional meaning the students intended to show, (2) perception of everyday school life, including social relationships and school performance, (3) support from guardians, classmates, and teachers, and (4) future visions and wishes for their future professional lives, including their interests and fears.

In the course of the interviews, the emojis were again used for deeper discussion and joint analysis of the photos. In this way, all children, regardless of their ability to abstract, their linguistic competencies, and whether or not they have SEN-L, to some extent, were able to take part in the participatory analysis of the photos, as prior research reported potential challenges when conducting the photovoice method with children having cognitive disabilities [45,46].

*2.2. Sample Description*

In this study, the data were collected in a multi-age classroom with different curricula in a Viennese inclusive school, although only students in the 4th grade of primary were considered. Due to the inclusive setting, the age range was higher than in regular 4th grade classes and included children from the ages of 11 to 12. The setting of the investigated class represents a specific and rather rare type of school concept, where the approach of "reverse inclusion" or "reverse integration" is applied. In this approach, students without SEN, who would normally attend a regular class, are included in a special education school, as opposed to the usual concept where students with SEN are integrated into regular classes [47]. At the time of the survey, there were 11 students in the class (of which 10 students participated in our project); 6 of these students had SEN-L and 3 students had no SEN and a regular curriculum, which in this case, were included in the special needs class regarding the concept of "reverse inclusion". The interviews were conducted during the piloting phase of the project, resulting in a sample size of nine students, in order to gain more detailed insights into the daily lives of students with and without SEN and possible new perspectives for future research time points. Interviews with all nine participants were conducted after the autonomous photovoice process. The mean age of the students was 11.33 years (SD = 0.5). Seven out of the ten students had an official diagnosis of special educational needs in learning (SEN-L), and one student additionally had a diagnosis of autism spectrum disorder (ASD), as overviewed in Table 1. Three of the interviewed students were taught according to the curriculum for regular students (RC, which indicates no SEN), and five were taught according to the special needs curriculum (SNC, which indicates SEN). One student was taught according to the curriculum regarding extensive support needs (SEF, which indicates SEN). The abbreviations SNC/RC/SEF indicate whether a student has SEN or not. Due to better readability, the terms have not been written out. The curricula differ in terms of the amount of learning content and the main learning objective. The special needs curriculum is a reduction of the regular curriculum, whereas the SEF focuses on learning and consolidating strategies for coping with everyday life. What needs to be emphasized in this context is the importance of the respective curricula for the further educational and life path of the students concerned. While students who are taught according to RC do not experience any disadvantages due to this, students with SNC and SEF are confronted with many barriers regarding their participation in the educational and labor market [48,49].

**Table 1.** Characteristics of the interviewed students.

|  | SEN | Gender | Curriculum |
|---|---|---|---|
| Student 1 | Yes | Male | SNC |
| Student 2 | No | Female | RC |
| Student 3 | Yes (additional information: ASD) | Male | SNC |
| Student 4 | Yes | Female | SNC |
| Student 5 | No | Female | RC |
| Student 6 | Yes | Male | SNC |
| Student 7 | No | Female | RC |
| Student 8 | Yes | Male | SNC |
| Student 9 | Yes | Female | SEF |

*2.3. Ethical Considerations*

The PATHWAY project was reviewed and approved by the ethical commission of the Educational Board of Vienna. Written informed consent was obtained from the students themselves as well as their legal guardians. Only if both signatures of the legal guardians and the child were obtained ethically, justifiable participation in the project and data collection procedure was ensured. In order to communicate the conditions of participation in a comprehensible way to all potential participants, the consent forms were distributed throughout the schools in different languages, as needed (Arabic,

Bosnian/Croatian/Serbian, English, Farsi, German, Hungarian, Polish, Russian, and Turkish). Participation could be withdrawn at any time during the project, even after its official completion. In this case, the data of the person concerned will be irrevocably removed from the data set. This has no consequences for affected individuals.

With regard to the methodological selection of photovoice, different support formats were installed in order to support students with and without SEN in the research process. "Photovoice methodology is employed to empower and transform the community through participation in the research process" [50]; to enable this participation, support material was designed (photo instructions and written instructions on how to use the camera on the task and the research process). Additionally, children could be supported in the implementation if they wished. None of the students took advantage of this offer. The conduction of research was carried out autonomously by all participants.

*2.4. Data Analysis*

The students' interview responses were analyzed using the MAXQDA software [51]. Based on the multifaceted concept of inclusion deriving from the quantitative approach of the Perceptions of Inclusion Questionnaire (PIQ), the main categories, including social inclusion, academic inclusion, and emotional inclusion, were constructed in a deductive approach [52]. Additionally, subcategories were added to the social inclusion category, which include peer relationship and teacher–student relationship. The data material were then coded according to these categories using MAXQDA. Due to the qualitative approach and to further specify the initial categories, the possibility of including additional categories inductively was left open and carried out within a second round of open coding. In doing so, a subcategory for social inclusion, namely, places of social interactions, was formed and developed from the data itself. To ensure the quality of the coding process, double coding was employed. The blind peer-analysis of the interviews showed an intercoder reliability of 0.85 (kappa) and 85.82% (agreement). With MAXQDA, the quantitative frequencies and contents of the coded statements were analyzed, and subsequently, meaningful statements were selected for a more detailed insight into the results.

## 3. Results

The social interactions category ($n = 146$), which can be further divided into three subcategories, was assigned the most statements from the data material. Most of the statements were classified into the peer relationship subcategory ($n = 63$), followed by teacher–student relationship ($n = 42$), and places of social interactions ($n = 41$). Secondly, a total of 60 statements were assigned to the academic inclusion category ($n = 60$). This category is followed by emotional inclusion ($n = 28$), with the fewest assignments.

*3.1. Social Inclusion*

3.1.1. Peer Relationship

In terms of social inclusion within their peer group, the students primarily reported positive experiences with their peers from class. Similar to the statements of **students 7 (RC), 4 (RC),** and **3 (SNC, ASD)**, **student 9 (SEF)** referred to the "happiness" emoji when referring to peer relationships in the context of school. In this regard, activities such as school excursions or spending time together, e.g., in the schoolyard during breaks, were particularly named.

> "[I am] happy when we go on a school trip together because we had a great time with all the classmates and then I was happy." (**Student 9, SEF**)

Additionally, regarding school activities, the students also described situations predominantly concerning private contexts and highlighted support provided by friends, for example, regarding private and not school-related issues. **Student 2 (RC)**, for instance, described how talking to friends from class when feeling sad has made her feel better afterward.

However, one student recounted an incident that caused anger due to a friend being insulted by another classmate. **Student 9 (SEF)** described that it is important to her to support her friends, to stand up for them, and to defend them from classmates who insult them, for instance.

> *"That student really caused so much anger to me that she bothered my friend. I always support my friends and I don't want her [another student] to do something to them that they don't like. They bother them, insult them or annoy them. I don't like that at all, I get really angry."* (**Student 7, RC**)

### 3.1.2. Teacher–Student Relationship

Based on the frequency of the nominations, it is seen that the students attach particular importance to the support and assistance provided by teachers in the everyday school context, as well as in private matters. The latter is mentioned mainly to the possibility of approaching the teacher for advice in the case of private issues. This support seems to be claimed by the students who have already known their teacher for a long time, and thus, have a long-standing teacher–student relationship.

> *"Yes, [I like] to play Uno with her [the teacher] so that I can talk about all the problems I have in my heart so that I can talk to her personally or explain to my teacher what is going on with me. [ . . . ] because I've known her for a long time. [ . . . ] And now it is important for me to contact the teacher to talk to me."* (**Student 9, SEF**)

In addition, the students report that their teachers provide important support in the academic context. According to the students' statements, a distinction can be made between the support related to the learning process, e.g., advice on learning strategies, and explaining subject content, e.g., introducing a new topic. **Student 7 (RC)** commented that the teachers assume an essential role in supporting the personal learning process and the academic school path in general.

> *"They are really very nice teachers. They want us to learn well. They have fun with it, they want everything to be good and if we ever need help, they want us to read first and try by ourselves. Trying ourselves first instead of just begging for help. They want us to do well and to have a good school experience."* (**Student 7, RC**)

However, other students claim to be dealing with their issues on their own and not accept or ask for any help or advice from the teacher. In the interviews, this view is mainly mentioned in the area of private and personal issues.

In contrast to the student statements mentioned above, **student 5 (RC)** described being sad and rather anxious when the teacher wants to talk in private.

> *"[I feel] sad, because, for example, when teacher 1 or teacher 2 says, you are coming with me for a moment I don't know what they want to talk about or where we are going. Then I'm also a bit scared and sad because I don't know what is going on. [ . . . ]"* (**Student 5, RC**)

### 3.1.3. Peer Relationship

During four of the conducted interviews, references were made to places of social interaction in school as a space to connect and interact with classmates. In this regard, the school's yard was mentioned frequently by the students. **Student 9 (SEF)**, for instance, described being very sad since the former schoolyard can no longer be used as before, as a container was placed there due to the enlargement of the school building.

> *"This is the sad smiley because we don't use the yard anymore and we did a lot together in this yard. We played, we played tag. We swung, we played hide and seek and yes, this yard is important to me. Now, the container school has appeared, and it has ruined everything for us."* (**Student 9, SEF**)

Additionally, **student 9 (SEF)** explained that the schoolyard likewise had an important part in the building and maintaining of social relationships and friendships, as it served as a place to talk with classmates in a private environment.

*"[ . . . ] because the yard was very important to us [ . . . ] and we told each other the whole truth and our secrets. We did all sorts of things in the yard, we had a lot of fun."* (**Student 9, SEF**)

In line with the other statements made during the interviews, **student 7 (RC)** also mentioned the yard and referred to a feeling of anger while thinking about the fact that they can no longer use it.

*"[ . . . ] I actually did [use] the anger smiley, because that makes me really angry. [ . . . ] I used it because of the new building in the yard where we can't play anymore. Yes, just because of the school, which needed more space, why does it have to be our yard?"* (**Student 7, RC**)

### 3.2. Emotional Inclusion

The students' emotional inclusion was dealt with mainly against the background of individual emotion regulation. The focus was on strategies as well as possibilities for emotion regulation in the school context.

Within the interviews, there are reports from the students of experiences in which emotion regulation is hardly successful and ends in negative states of tension. This is, for example, explicitly reported by **student 5 (RC)**, who enters a negative reaction cascade in stressful situations. When asked whether sharing her negative feelings with friends makes her feel better, **student 5** reported that this only increases her anxiety.

*"My mother knows about it. For example when I have felt sick and vomited, I always get panic attacks and often pull my hair and bang my head on the floor because I'm so scared."* (**Student 5, RC**)

The interview guidelines encompass questions about whether the students can consult someone at school, e.g., classmates or teachers, to talk about their feelings. Some of the students report talking to both school friends and teachers about their feelings and problems. Like other participants, **student 9 (SEF)** reports turning to her teacher for help. As in other interviews, e.g., **student 2 (RC)** said that she talks to her friends when she has problems and feels better afterward. Other students, however, say that they never speak to others about their feelings at school, including **student 1 (SNC)** and **student 5 (RC)**.

In addition to reports about not sharing one's feelings and problems with others, some students report hiding their feelings altogether. Among other statements, **student 7 (RC)** explained that she hides her anger from her classmates and teachers as well as she can, and deals with it purely within herself.

*"I hide my anger a little bit because I don't want others to see that I just don't look good like this, that I don't look happy. [ . . . ] I don't want anyone to see that I'm angry. I hide my anger inside. [ . . . ] I do it in such a way that I simply say, great, it doesn't work. It is better to control my anger through my self-affirmations than to show it on my face."* (**Student 7, RC**)

### 3.3. Academic Inclusion

As shown in the theory, during the interviews, the students' statements regarding academic inclusion were mainly related to academic self-concept. Some of the students' statements imply that the self-perception of their academic self-concept differs when it comes to different school subjects, and therefore, applicable for some participants, cannot be defined as a global construct, but rather a domain-specific one. Mathematics is most often referred to as a subject in which the students think they are not good at. For example, **student 2 (RC)** associated Math with the two emojis displaying "anger" and "disgust". **Student 9 (SEF)** wished to be able to not make mistakes in Mathematics but has had

bad experiences so far, which resulted in a negative self-concept regarding this specific school subject.

> *"Yes, because I'm always worried when I get a bad grade or I'm afraid of doing Math that I won't understand it the right way and kids will laugh at me. I understand Math anyway, but I don't understand the outcome of calculations. I don't want to make mistakes. I've already made a lot of mistakes in my life [ . . . ]."* (**Student 9, SEF**)

In addition, other students recount differentiated subject-specific perceptions of their own academic self-concept and state that they see themselves as good students in different subjects, e.g., in English or German. In the context of academic performance, such as doing homework or exams, the students' perceptions vary. **Student 4 (SNC)** stated that she is very excited about exams and assigned the emoji displaying "happiness" to them. On the other hand, some students state that they believe that difficult performance situations will probably not be mastered, and that these trigger negative emotional states.

> *"So I can do a few homework exercises, but I'm a little desperate [ . . . ] I already know English well [ . . . ] but I'm afraid that I'll say something wrong. [ . . . ] Therefore the desperate emoji."* (**Student 1, SNC**)

Like **student 1 (SNC)**, **student 9 (SEF)** also reported being anxious, but in addition to homework, the statement furthermore includes assignments and exercises at school mainly concerning the school subjects German and Mathematics.

> *"I'm always afraid of schoolwork because I can't figure out what I should or shouldn't do. I'm always scared most of the time because Math and German are important and that's why I'm always scared because I'm always sitting at my desk doing Math or German or something and thinking about the right way to solve the tasks. You must practice, practice, practice every day and that's good for life, but still, I'm always afraid of schoolwork because I can't get it right."* (**Student 9, SEF**)

**Student 6 (SNC)** also reported being afraid, but in addition to the fear of doing homework, he also referred to the fear of not receiving good grades.

> *"If I can't do homework well, then I'm afraid. Then I'm afraid because I might not get a good grade."* (**Student 6, SNC**)

In addition to these experienced negative emotional states, some students report that failure at school leads to the reinforcement of emotions through the impact of peer relationships. As in other interviews, **student 5 (RC)** stated that he is afraid of being laughed at by his classmates, which in turn, leads to an increase in his state of tension.

> *"That I'm a bit nervous there. When I don't get as many points as I am expected to receive. Because then many people in class laugh at me because I didn't get as many points as the others. Then they laugh at me sometimes."* (**Student 5, RC**)

### 3.4. Perceptions of Inclusion of Students at the Intrapersonal Level

Table 2 shows the results regarding the perception of inclusion of students at the intrapersonal level in the context of social, emotional, and academic inclusion. **Student 1 (SNC)** does not mention any statements related to social inclusion, but shows a positive perception regarding emotional inclusion. In terms of academic inclusion, positively and negatively connoted perceptions regarding the different school subjects, as seen in **student 3 (SNC, ASD)**, can be observed. **Student 2 (RC)** and **student 9 (SEF)** demonstrate a positive perception in the context of social and emotional inclusion. Regarding academic inclusion, both reveal a negative perception, related exclusively to the subject of Mathematics. Similarly, **student 5 (RC)** describes a negative academic self-concept that is predominantly related to Mathematics, though a negative perception can also be observed in the context of social and emotional inclusion. **Students 3 (SNC, ASD), 4 (SNC)**, and **7 (RC)** show a positive perception in the context of social inclusion. While **students 3 (SNC, ASD)** and **4 (SNC)** only comment peripherally on emotional inclusion, **student 7 (RC)** prefers to deal

with their emotions themselves and shows, the same as **student 4 (SNC)**, a positive academic self-concept. Neither **student 6 (SNC)** nor **student 8 (SNC)** address the perception of emotional inclusion. **Student 6's (SNC)** statements show a negative perception of social as well as academic inclusion. **Student 8 (SNC),** on the other hand, describes a positive academic self-concept but does not refer to social inclusion at all. As can be seen in Table 2, no clear differences can be shown between the students with and without SEN, as the results differ individually.

**Table 2.** Perceptions of inclusion of students at the intrapersonal level of the interviewed students.

| S | SEN | G | C | Social Inclusion | Emotional Inclusion | Academic Inclusion |
|---|---|---|---|---|---|---|
| **S1** | Yes | M | SNC | Not mentioned | No support is desired in emotionally challenging situations from teachers or classmates | Negative general ASC Positive ASC in E, but also afraid to fail |
| **S2** | No | F | RC | Positive general perception High confidence level with teachers, strong class involvement, seeking support in school and private contexts | Seeking support in emotionally challenging situations from teachers or classmates | Negative ASC in Math |
| **S3** | Yes (ASD) | M | SNC | Positive general perception High confidence level with friends, strong class involvement | Ascribes little importance to emotions, indifferent observation | Positive ASC in Math, Ger, Sports Negative ASC in E |
| **S4** | Yes | F | SNC | Positive general perception Strong class involvement | Not mentioned | Positive general ASC |
| **S5** | No | F | RC | Negatively connoted perception Low confidence level with teachers and classmates, not seeking support in school nor private contexts | Negative states of tension, negative reaction cascade in stressful situations, no support desired in emotionally challenging situations | Negative ASC, fear of being laughed at by peers, specifically Math |
| **S6** | Yes | M | SNC | Negatively connoted perception Teacher's support in the school context, no communication with teachers or classmates about private issues | Not mentioned | Negative general ASC |
| **S7** | No | F | RC | Positive general perception Importance of friendships, sense of justice toward friends, high confidence level with teachers | Positive general perception Importance of friendships, sense of justice toward friends, high confidence level with teachers | Positive general ASC |
| **S8** | Yes | M | SNC | Not mentioned | Not mentioned | Positive general ASC, referring to E, Ger, and Math |
| **S9** | Yes | F | SEF | Positive general perception Supportive behaviors, conflict resolution, high confidence level with teachers, strong class involvement | Seeking support in emotionally challenging situations from teachers or classmates, possesses strategies for emotion regulation | Negative ASC in Math |

Notes: Abbreviations: S = student; SEN = special educational needs; G = gender; M = male; F = female; ASD = autism spectrum disorder; C = curriculum; SNC = special needs curriculum; SEF = support needs; RC = regular curriculum; E = English; ASC = academic self-concept; Math = Mathematics; Ger = German.

## 4. Discussion

This study aims to provide insights into students' perceptions of their everyday school lives by breaking it down to their perceptions of academic, social, and emotional inclusion. To access the experience of students with and without SEN, the photovoice method and the use of emojis as representors of emotional expressions were chosen.

The results show that there is little to no difference between the groups of the students with and without SEN in their evaluation of the emotional impact and associations of different school experiences, but that the spectrum of positive and negative evaluation is broadly distributed within the sample group as a whole.

Regarding the students' perception of social participation, both students with and without SEN report positive and negative experiences concerning their peer relations in class. However, this finding does not support previously presented quantitative studies which imply that students with SEN are more likely to experience low participation and fewer friendships than their peers without SEN (e.g., [21–23]). Despite the broad consensus of quantitative findings on the social disadvantage of students with SEN, recent research findings indicate that there are no significant differences in social participation among the students of a class [53]. With regard to the present study, it should be noted that all participating students attended the same class, and thus, had the same network of classmates. The research field displays a specific design of inclusive education, which involves the "reverse inclusion" or "reverse integration" approach. The concept of "reverse inclusion" implies that students without SEN and without disabilities, who would attend regular schools and classrooms in the ordinary case, attend school in the educational setting of special education schools [47]. This school setting, which is basically characterized by special education characteristics but also includes occasional reversed inclusion classes, makes it clear that the role of the "to-be-included" in this setting is ascribed to a different group of students than usual. In the present case, it is regular school students who are included in an existing system for people otherwise defined as "different" [54,55]. Due to the fundamentally different schooling conditions and approaches to different educational needs in students compared to regular school settings, the results showing no clear differences between students with and without SEN in terms of social inclusion regarding peer interactions seem plausible.

With regard to the social relationship with teachers, the students of the investigated class have different experiences. One student reports about actively approaching the teacher and having joint (relieving) conversations. What seems striking in the statement of student 9 (SEF) is the emphasis on the student's own active part in initiating conversations with the teacher ("*it is important to me to contact the teacher to talk to me*"). This statement leaves open whether conversations to exchange problems are instigated by the teacher or whether the student's own initiative and request are fundamentally decisive here. What is also striking about this statement is that when the teacher is actively approached, the student does not only want to talk about formal, school-related content, but also about topics that go beyond the school context ("*all the problems I have in my heart*"). In this case, the student–teacher relationship is characterized by a basis of trust on the part of the student, which makes it possible to talk about different emotionally charged topics. In contrast, we have statements of the students who have an insecure relationship pattern with their teachers and seem to be in a persistent state of alarm. For student 5, especially, there is no evidence here of basic trust in teachers, but rather a basic distrust and skepticism. In her statement, alertness refers primarily to the student's fear (accompanied by the emotions of sadness) of being reprimanded or receiving information about poor school performance. The negatively connoted feelings associated with one-on-one conversations with teachers thus refer exclusively to potentially school-related conversation content. In accordance with the present results, previous studies demonstrate that student–teacher relationships can be influenced by the amount and intensity of informal conversations [56,57]. Positive student–teacher relations, and therefore, higher social inclusion into the class are considered to be characterized by both teachers and students being proactive when it comes to initiating

informal conversations and contact, but also by talking about a broad range of formal and informal topics with each other [57]. According to the literature, this result especially applies to classrooms characterized by a high degree of heterogeneity and diversity of student needs [58].

Another important finding was that four of the nine interviewed students mentioned the schoolyard as highly important for social interactions with both peers and teachers. The schoolyard as an important place of social interaction is characterized by the students as a location outside of the school building and the classroom, and so there is a distance from the academic learning and requirement space whereby a demarcation to the academic learning and requirement space can be established, and informal learning takes place [59]. According to the students' statements, the schoolyard is associated with more freedom and recess. In line with this finding, previous studies evaluating the impact of context and places of interactions on the perception of social participation and connectedness also show that not only conversations about informal topics, but also within informal contexts and outside of the classrooms, have positively influenced social relationships in the school context [56–58]. Against the background of the findings of the current and previous studies, it can be interpreted that the schoolyard is associated by the students as a form of a safe space that can be independently chosen, co-constructed, and designed. With that in mind, regarding the inclusive design of educational settings promoting social inclusion, the question arises as to which spaces must, can, and should be created that lead to inclusion and educational processes at non-formal and informal levels. In the light of these results, it therefore seems interesting to explore the role of the schoolyard as a potential learning space, and to check for possibilities of alternative space used to create innovative pedagogical settings [60].

Talking about innovative, inclusive learning opportunities, the positive recognition by one student of the teachers' implementation of inclusive teaching features also seems noteworthy. This finding is consistent with that of Lindner et al. [20], who showed that students mainly focus on relationships when it comes to their perceptions of social inclusion, but highlights mutual support between peers and teachers and an associated supportive network as decisive for promoting social inclusion. Interestingly, within the aforementioned study, the inclusive elements that the students describe in terms of instructional design (promoting fun when learning) were mainly the parents' mentions. The parents of both students with and without SEN wanted the teachers to use inclusive teaching and learning methods (e.g., "*creating a nice/pleasurable atmosphere*", "*providing easier tasks*", "*individual support*", and "*creating a pleasant climate*"; Lindner et al. [20] (p. 5)).

Referring to the students' perception of their emotional inclusion in school, it is notable that the students do not mention their display of emotions in the classroom, or their emotional associations with going to school and everyday school life, per se, independently of the use of emojis in the photovoice method, which should contribute to the collection of core experiences in everyday school life. However, what the students were very concerned about, and what was mentioned in the interviews, were the self-regulation processes to deal with emotions in everyday school life. In this context, emotion regulation is defined as "*the capacity to modulate one's emotional arousal such that an optimal level of engagement with one's environment is fostered*" [61] (p. 907); encompassing competencies to recognize and identify self-experienced emotions, as well as to use regulation strategies to manage these emotional states, can result in internalizing as well as externalizing behavioral symptoms at an intra- and interpersonal level [62–64]. What derives from the interviews with the students is information about internalizing as well as externalizing regulative behavior. The students with internalizing regulation strategies describe that they do not channel negative feelings outward but keep them to themselves. Negatively connoted feelings are assumed to be bad for others, and therefore must be hidden, as they do not promote a good image of oneself. Only positive feelings are considered to be openly desired feelings. In this context, it remains unclear by whom this division into desired and undesired feelings is made and for which settings this categorization is applicable. Up until now, far too little

attention has been paid to students' emotion regulation and coping strategies in previous research. In most cases where research was conducted on students' emotion regulation, it is associated with pathologized characteristics or disorders (ADHD, e.g., [65,66]; ASD, e.g., [67–69]). However, there is a growing body of the research literature dealing with emotion regulation and the emotional display of teachers [70–73].

With regard to academic inclusion, the students mainly express perceptions that refer to their academic self-concept. The results show that the students' self-perception of their academic self-concept does not differ between students without and with SEN. It is noticeable that within their statements about academic self-concept, the students mainly implicitly refer to different school subjects or differentiate their academic self-concept explicitly according to the individual school subjects. For example, student 3 (SNC) shows a negative academic self-concept regarding English competencies, but reports being successful in Mathematics and German. What was mentioned in all the students' statements about their academic self-concept was very strong pressure to perform and increased expectations (self and external expectation) that come with the fear of not being able to meet them. In this context, the students' statements give rise to the interpretation that their subject-specific or general negative self-concept are historically and biographically shaped and transferred from formal learning experiences to other areas of the students' lives, such as basic life management or vice versa (e.g., "*I've already made a lot of mistakes in my life*"). The distribution of positive and negative tendencies regarding the students' academic self-concept is balanced concerning the subdivision into students with and without SEN. Accordingly, no difference in the reports on subject-specific academic self-concept between the groups of students (SEN and no SEN) was found. This outcome is contrary to previous studies that found that learners with SEN are at risk for having more negative subject-specific academic self-concept than their peers without SEN [74,75].

Thus, to outline the students' perception of inclusion, a closer look into their perceptions of inclusive domains adapted from the PIQ (social, emotional, academic inclusion) seems noteworthy on an intrapersonal level, as all domains are highly influential for the students' school inclusion, apart from a focus solely on learning [4–6,30]. In past quantitative studies, there were intrapersonal differences regarding the perceived levels of inclusion within the three dimensions. However, the rankings of the three domains regarding the perceived levels of inclusion vary across the studies without revealing a consistent pattern for students with and without SEN [30,32,74,76].

The results of current studies show that the intrapersonal perception of social, emotional, and academic inclusion does not provide a consistent picture across all domains and among all students (for example, positively connoted emotional inclusion and negatively connoted academic inclusion perceived by the same student). Furthermore, none of the students showed a consistently intrapersonal positively connoted perception of social, emotional, and academic inclusion. For student 4 (RC) and student 8 (SNC), there is positive attribution in all the areas mentioned, but not all dimensions of inclusion are addressed, so no conclusion can be drawn about a fundamentally positive perception of inclusion. Based on these results, it is not possible to draw any conclusion about an all-embracing positive inclusion of one of the interviewed students, which is, in fact, the goal of the inclusive classroom setting and seems to be missed in this case. Conversely, student 5 (SNC) exhibits a consistently negative connotation of all inclusion domains, resulting in a high risk of exclusion. Students 4 (RC) and 7 (RC) both display a positively connotative perception in terms of social and academic inclusion. Social inclusion, and subsequently, social interactions functioning as recovery periods between learning activities may lead to better academic performance [57], which in turn, may affect academic inclusion perceptions. However, with respect to students 4 (RC) and 7 (RC), no clear relationship between social and academic inclusion can be checked here. Previous research provides insights into topic-relevant findings, stating that a negative self-concept can function as a predictor of loneliness, and therefore, a lack of social inclusion for students with SEN in an inclusive classroom setting [77]. In this context, it is worth investigating the relationship between

academic self-concept and social inclusion in the future, as there is now a stronger focus on the connection between academic performance and social participation in school [53,78,79].

Regarding student 7 (RC), no explicit references to emotional inclusion are provided, whereas a self-regulated suppression of feelings is reported. This suppression could indicate past negative experiences with the expression of feelings or internalized regulation strategies. Based on this, the emotional inclusion of student 7 cannot be categorized as consistently positive. In general, a negative regulation competence of feelings can be associated with reduced school inclusion [64], although no connection between this and the perception of the emotional inclusion of student 7 (RC) can be derived from the qualitative data collected within the current study. As summarized by Kopelman-Rubin et al. [64], associations between emotion regulation and emotional well-being [80], social competence [81], and academic achievement [82] have been found in the past. Considering the three levels of the PIQ, understanding the emotion regulation of students with SEN from an inclusion perspective takes on a greater significance and opens an important area of research for the implementation of inclusive education. Concerning student 3's (SNC; ASD) perception of the inclusion domains, it is not possible to make a precise statement for neither emotional nor social inclusion, as different statements are made within the subdomains. Previous research on people diagnosed as having ASD refute the assumption that this population group does not have a general deficit in the perception and understanding of emotions [83–85], but rather has difficulties in classifying and describing their own emotions [86]. To what extent this is transferable to student 3 remains open, but provides a starting point for future photovoice interviews with students with ASD [46] regarding their perception and verbalization of their feelings.

## 5. Conclusions

Overall, several conclusions can be drawn from this study. First, in this study, there are little to no differences in the perception of inclusion between students with and without SEN, which may be related to the inclusive classroom setting of the investigated research field. Social inclusion, including student–teacher relationships and peer relationships, is perceived differently by students attending the same classroom. However, almost all students mentioned the schoolyard as an important social space, suggesting future research on how social spaces within the school context which are not associated with formal learning situations can influence general social inclusion in school. Concerning emotional inclusion, it should be noted that the students were very concerned about self-regulation processes to deal with emotions in everyday school life. Since there are still few studies in this area that are not related to specific disorders or pathologized characteristics, this is an important area for future research. Another noteworthy aspect is the inconsistent results of the students' intrapersonal perception of the three dimensions of inclusion, namely, social, emotional, and academic inclusion, which, in turn, provides an opportunity for further studies to explore possible relationships between these domains and a general perception of students' inclusion and possible implementation for improving these.

**Author Contributions:** Conceptualization, A.P., J.H. and K.-T.L.; methodology, A.P., J.H. and K.-T.L.; software, A.P. and J.H.; validation, A.P., J.H. and K.-T.L.; formal analysis, A.P., J.H. and K.-T.L.; investigation, A.P., J.H. and K.-T.L.; resources, A.P., J.H. and K.-T.L.; data curation, A.P., J.H. and K.-T.L.; writing—original draft preparation, A.P., J.H. and K.-T.L.; writing—review and editing, A.P., J.H. and K.-T.L.; supervision, A.P., J.H. and K.-T.L.; project administration, A.P., J.H. and K.-T.L.; funding acquisition, K.-T.L. All authors have read and agreed to the published version of the manuscript.

**Funding:** This research was funded by Oesterreichische Nationalbank grant number 18567 and The APC was funded by Open Access Funding by the University of Vienna.

**Institutional Review Board Statement:** The PATHWAY project has been reviewed and approved by the ethical commission of the Educational Board of Vienna.

**Informed Consent Statement:** Informed consent was obtained from all subjects involved in the study.

**Data Availability Statement:** The data presented in this study are available on request from the corresponding author. The data are not publicly available due to the ongoing project and the protection of the anonymity of the participants.

**Conflicts of Interest:** The authors declare no conflict of interest.

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
