# Peer review of "Investigating the Self-Perception of Social, Emotional, and Academic Inclusion of Students with and without Special Educational Needs through Photovoice"

_education, doi:10.3390/educsci13040423_

Round 1

Reviewer 1 Report

Congratulation to the study authors. The text is well constructed, and the research design is of high quality. I want to improve the study with my additional comments. 

The authors write about SEN students' social, emotional, and academic inclusion in the introduction. This part is too general. SEN is a very wide category. Please rewrite this part and highlight which authors and which type of SEN have been discussed, for example, in the case of Mariano et al. I would appreciate it if you expanded this part with these points. A theoretical background must emphasize the previous research relating to SEN-L if the study is focused on this phenomenon.

You are writing: In this context, data collection especially highlights the extent to which the phenomenon of "learning disabilities" must be seen as socially constructed and discussed in terms of inherited educational poverty. What do you mean by this? I would appreciate it if you could clarify and support it with literature. Reflection is needed later when the social background is in focus.  

Your research was conducted in a reverse inclusion class/school. Please, provide more about this school-type because it is not so general. 

Please write more about the qualitative analyses methodology. (For example, you used software during the analyses, etc.).

It is necessary to explain the acronyms: RC, SNC, and SEF. What does it mean in your educational system? There aren't widely used concepts. 

The number of samples is low. You have to justify why only nine students took part in the study. It would be helpful if you could describe the school and the class in more detail (size, average SES, number of school support professionals, and other conditions). A more thorough presentation of the interviewees would also need necessary. 

3.1.2 and 3.1.3. has the same title. 

One limitation of your research is the sample size, and the reserve inclusion class is another. The quantitative literature can be used to discuss both of them since they were conducted under different circumstances. So please write more carefully. 

Reviewer 2 Report

Dear Author(s),

Many thanks for giving me the opportunity to read you interesting article. It is a very worthwhile and informative article. I am including feedback below, one of my main concerns is the lack of an in-depth discussion in terms of the ethical considerations of the study particularly since the study was undertaken with minor and with students with SEN. I think that it is of utmost importance that you include this content in the redraft.  Kind regards, Reviewer

Abstract

Line 9-11: this line is unclear in terms of structure/meaning. Can you rephrase? I think it is best to keep the language simple and understandable for accessibility for all readers.

Line 19-20: rephrase, e.g., a contrast between…. was not detected.

Line 20 -21: I am not sure what this sentence means - The emotional inclusion of the students was addressed primarily in the context of individual emotion regulation.

Introduction

In the introduction section I suggest you present more information about your study before you present the literature review. Some of the content from the Data/Methods section might be good to move up here as it would explain what the study is about before the reader engages in the literature review.

Line 37: there is a quote in the wrong format

Line 43-46:  the content in these sentences needs to be explained further for the reader, e.g., what do you mean by verifiable on a quantitative level? What do you mean by - whether the three dimensions are autonomously named by the participating students?

Section 1.1: it might be a good idea to give the reader a little bit more information about the questionnaire and how it influenced the study.

Line 54: what is a.o.? tell the reader. This line in general is short and does not make sense.

Line 57: Can you give more context? Tell the reader why ‘Some studies, such as McCoy and Banks [11] and Skrzypiec et al. [12], report lower emotional inclusion scores for students with SEN,’

Line 59: why is emotional regulation important for inclusion? Tell the reader.

Line 70: I suggest you delete the word especially

Line 74: You use special educational needs, you have used SEN previously, be consistent in terms of using the abbreviations you introduce. Please double check whole article for consistency.

Section 1.5: The present study, I would like to know more about the present study at the beginning of the article before I read the literature reviewed. I found that I was reading the literature and I was not sure what the point of the literature I was reading was.

Line 104: You say: special educational needs and to identify important issues for the children that may be overlooked by adults, interviews were conducted using the photovoice method. What was the rationale for this? What do you mean by overlooked by adults? It might be good to add a short discussion on student voice here and its benefits etc.

Line 105: Provide reference for photovoice method literature

Line 106: What do you mean by important topics? Explain

Line 110: What is SEN-L?

Line 123 -125: These sentences are very clear and show the purpose of the study, it would be good to have some description like this at the beginning of the paper so that the read knows more about the study.

Line 128: Give the age ranges for 4th grade of primary school, e.g., age 9-10 and also the age range for transition to work life

Data/Methods Section: It would be best to present the methods used before you present the participant profiles. Present information to the reader on the PIQ questionnaire (how many items etc.) and also the theory/literature around the photovoice method first. Why were there only 3 students without SEN included whilst there were 6 with SEN? What was the rationale for this? How might it impact on the findings?

You are also missing an in-depth discussion on the ethical considerations of the study – where did you get ethical approval from? What ethical considerations were there in terms of working with children, particularly those with SEN? What protocols did you introduce? Who gave consent? How were the participants informed about the study? What process did you use to ensure understanding of participants? How would you know if they felt uncomfortable and wanted to withdraw?

Line 141: What do you mean by special educational needs in learning? Do these students have an official diagnosis? What were there learning difficulties specifically? Could you add this information to table 1?

You have two sections (3.1.2 and 3.1.3) which have the same heading – if the content relates to the one theme it is best to just have one section but I think from reading the article you need to change the heading of 3.1.3

Results section – In the presentation of the data in the results section, it is unclear to me which students present with SEN and which students do not. It would be good to provide a clearer comparison of findings between the two groups you studied. Also only 3 of the students did not present with SEN, whilst 6 did. How do you think that this impacted on your findings? There are a lot of direct quotes from the same students (e.g., 5/7/9) might you consider providing more variety in terms of quotes from other participants?

Line 386: Please check line for clarity/meaning

Line 391 – 421: You are introducing contextual details that would benefit the reader earlier in the article.

Line 546: Incorrect referencing style used.

Discussion section: There is some repetition in this section in terms of the findings already presented in the findings section. It would be good to read more about the significance of your study? Why is it important? What does it add to the research field? What should happen next? I think that this section would also benefit from more references to the literature to reinforce points made.

I think that the literature review section could be strengthened by more detail in terms of the areas chosen by students where they felt good etc. This would help you then in the discussion section to make more comparisons to the literature.

Suggested readings -

Gaona, C., Palikara, O., and Castro, S. (2019) ‘I’m Ready for a New Chapter’: The Voices of Young People with Autism Spectrum Disorder in Transition to Post-16 Education and Employment. British Educational Research Journal, Vol. 45(2), 340-355. doi:10.1002/berj.3497.

Norwich, B., and Kelly, N. (2004) Pupils’ Views on Inclusion: Moderate Learning Difficulties and Bullying in Mainstream and Special Schools. British Educational Research Journal, 30 (1), pp.43-65.

Prunty, A., Dupont, M., and McDaid, R. (2012) Voices of students with special educational needs (SEN): views on schooling. Support for Learning, Vol.27: 29- 36. doi:10.1111/j.1467-9604.2011.01507.x

Riley, K. (2004) Voices of Disaffected Students: Implications for Policy and Practice. British Journal of Educational Studies, Vol. 52(2), pp. 166-179.

Sellman, E. (2009) Lessons learned: Student Voice at a School for Pupils Experiencing Social, Emotional, and Behavioural Difficulties. Emotional and Behavioural Diffculties, Vol. 14(1), pp. 33-48.

Squires, G., Kalambouka, A., and Bragg, J. (2016) A Study of the Experiences of Post Primary Students with Special Educational Needs. Trim: National Council for Special Education. http://ncse.ie/wp-content/uploads/2016/07/NCSE-AStudy-of-the-Experiences-of-Post-Primary-Students-with-Special-Ed-Needs.pdf (accessed 7th November 2020)

Round 2

Reviewer 2 Report

Dear Author(s),

Many thanks for addressing the comments that I made. It is greatly appreciated and I feel that the changes you have made to the paper add to it. You have done great work. Upon reading the article again I picked up on these very minor issues. Best of luck with the publication, Reviewer 2

Line 32 – 35:  You say ‘In order to map a perception of inclusion not only at the level of SEN approval, but an in-depth approach through three different levels (emotional, social, and academic inclusion) was also applied to map the thoughts and feelings of all participating students regarding their perceived inclusion.’  This sentence needs to be redrafted as it does not make sense. E.g., An in-depth approach was utilised that investigated three different levels of inclusion, emotional, social, and academic. This was done in order to….

Line 66: SEN is a broad term rather than category?

Line 66/67: ‘while the spectrum of SEN in one country is broad as well, as it is also differentiated according to the type of impairment.’ Can you rephrase this? E.g., each country categorises SEN differently due to differences in diagnostic testing and criteria.  Is this what you mean?

Line 173: You use the term ‘limited power’ is this what you mean or do you mean something like limited language/linguistic ability?

Line 237: do you mean ensured rather than ensure?

Line 254: provide reference for where we can read more about the software please.

Author Response

Thanks a lot for your comments! Please see the attachement.
